# Transcriptome-Wide Analysis of Low-Concentration Exposure to Bisphenol A, S, and F in Prostate Cancer Cells

**DOI:** 10.3390/ijms24119462

**Published:** 2023-05-30

**Authors:** Sergio A. Cortés-Ramírez, Ana M. Salazar, Monserrat Sordo, Patricia Ostrosky-Wegman, Miguel Morales-Pacheco, Marian Cruz-Burgos, Alberto Losada-García, Griselda Rodríguez-Martínez, Imelda González-Ramírez, Karla Vazquez-Santillan, Vanessa González-Covarrubias, Vilma Maldonado-Lagunas, Mauricio Rodríguez-Dorantes

**Affiliations:** 1Laboratorio de Oncogenómica, Instituto Nacional de Medicina Genómica, Mexico City 14610, Mexico; sergio.cortesram@gmail.com (S.A.C.-R.); mp.miguelmorales@gmail.com (M.M.-P.); marian.cruz.bqd14@outlook.com (M.C.-B.); correo.garciabeto@gmail.com (A.L.-G.); griseldargzmtz123@gmail.com (G.R.-M.); 2Instituto de Investigaciones Biomédicas, Universidad Nacional Autónoma de México (UNAM), Ciudad Universitaria, Mexico City 70228, Mexico; anamsm@biomedicas.unam.mx (A.M.S.); ostrosky@iibiomedicas.unam.mx (P.O.-W.); 3Departamento de Atención a la Salud, Universidad Autónoma Metropolitana-Xochimilco, Mexico City 04960, Mexico; imeldagr14@gmail.com; 4Laboratorio de Innovación en Medicina de Precisión, Instituto Nacional de Medicina Genómica, Mexico City 14610, Mexico; kivazquez@inmegen.gob.mx; 5Laboratorio de Farmacogenómica, Instituto Nacional de Medicina Genómica (INMEGEN), Mexico City 14610, Mexico; vgonzalez@inmegen.gob.mx; 6Laboratorio de Epigenética, Instituto Nacional de Medicina Genómica, Mexico City 14610, Mexico

**Keywords:** prostate cancer, endocrine disrupting chemicals, transcriptome, bioinformatics, DNA damage

## Abstract

Bisphenol A (BPA) is a ubiquitous synthetic compound used as a monomer in the production of polycarbonate plastics and epoxy resins. Even at low doses, BPA has been associated with the molecular progression of diseases such as obesity, metabolic syndrome, and hormone-regulated cancers due to its activity as an endocrine-disrupting chemical (EDC). Consequently, the use of BPA has been regulated worldwide by different health agencies. BPA structural analogs such as bisphenol S and bisphenol F (BPS and BPF) have emerged as industrial alternatives, but their biological activity in the molecular progression of cancer remains unclear. Prostate cancer (PCa) is a hormone-dependent cancer, and the role of BPA structural analogs in PCa progression is still undescribed. In this work, we use an in vitro model to characterize the transcriptomic effect of low-concentration exposure to bisphenol A, S, or F in the two main stages of the disease: androgen dependency (LNCaP) and resistance (PC-3). Our findings demonstrated that the low concentration exposure to each bisphenol induced a differential effect over PCa cell lines, which marks the relevance of studying the effect of EDC compounds through all the stages of the disease.

## 1. Introduction

Bisphenol A (BPA) is the monomer used in the synthesis of epoxy resins and polycarbonates, also used as a plasticizer for thermoplastic polymers [1]. Global production of BPA increased from 5 million metric tons to 8 million between 2012 and 2016, and it is expected to reach 10.6 million metric tons by the end of 2022 [2]. On the one hand, BPA derivates are used in a wide broad of applications, including polycarbonates for bottles, toys, thermal paper, household appliances, and medical equipment. On the other hand, epoxy resins are applied as protective coatings for food and beverage packaging, adhesives, paints, and electronic laminates [3]. Due to its extensive number of uses, BPA residues have been found in agricultural and industrial soils [4], air [5,6], water, sewage, sediments [7], and food [8]. Hence BPA is a ubiquitous molecule, and humans are constantly exposed to it through multiple routes such as food, the skin, particle ingestion, and inhalation [9]. Health agencies have determined the presence of BPA in more than 90% of urine samples in a human reference population (US population) [10]. However, current evidence regarding the potential harm of BPA exposure to human health remains inconclusive, leaving uncertainty regarding the specific conditions under which it may be harmful to human health.

BPA exposure is involved in different human diseases due to its activity as an endocrine disruptor chemical (EDC). According to the U.S. Environmental Protection Agency (EPA), endocrine disruptor chemicals are defined as exogenous agents capable of interfering with hormone action in their synthesis, transport, binding, and metabolism, causing adverse effects in an organism or population [11]. EDCs are capable of disrupting hormone receptors at their rapid genomic and non-genomic signaling. In 2019, the Expert Consensus Statement identified ten different molecular mechanisms underlying EDCs action, including hormone receptor agonism, hormone receptor antagonism, signal transduction, regulation of hormone receptors expression, epigenetic modifications, hormone synthesis regulation, hormone transport, distribution, and clearance, as well as cell fate dysregulation. It is noteworthy that BPA is capable of exerting its EDC activity by using nine of these ten molecular mechanisms [12]. Thus, it is not surprising that BPA has been related to several human diseases such as obesity [9,13], cardiovascular disease [14], type 2 diabetes [15], polycystic ovarium syndrome [16], asthma [17], as well as in the progression of hormone-regulated cancers such as breast [18], testicular [19], thyroid [20,21] ovary [22,23], and prostate [24,25]. Although there is evidence linking BPA exposure to different diseases, the controversy surrounding the biological effect of EDCs remains. This is because most studies testing these compounds, including BPA, have been used in higher concentrations than those found in the environment. To solve this conundrum, the U.S. Environmental Protection Agency (EPA) and the National Toxicology Program (NTP) defined low-dose effects as the biological changes that occur at (1) the range of typical human exposures, (2) doses at lower ranges than those used in standard protocols (below traditional toxicological assessment), and (3) any dose below lowest observed adverse effect level (LOAEL). As a result, it is important to test low concentrations of EDCs because of their biological relevance. Moreover, another reason for characterizing the biological impact of low-concentration EDCs is the presence of their broad mechanisms, which exhibit non-linear dose-response effects, commonly known as non-monotonic effects (NME) [26]. Consequently, the effects of EDCs at low concentrations are difficult to predict and must be characterized.

The European Food Safety Authority (EFSA) reduced the BPA reference dose, prompting restrictions and regulations regarding its use and production in North America and the European Union [27,28,29]. Governmental institutions and public awareness led to the production and utilization of structural analogs, such as bisphenol F (BPF; 4,4′-methylenediphenol) and bisphenol S (BPS; 4-hydroxyphenyl sulfone), which have shown similar cytotoxic and endocrine disruptive activity [30,31]. Therefore, it is pertinent to fully characterize the molecular role of low-dose exposure to BPA and its structural analogs, BPS and BPF, in several disease states. Here, we will focus on the investigation of the effect of the low dose of BPA, BPF, and BPS in prostate cancer, a hormone-regulated cancer.

Prostate cancer (PCa) is the second leading cause of cancer among men worldwide, and it is the main cause of cancer in Western countries [32]. Prostate gland organogenesis is highly regulated by the presence of steroid hormones, where androgens and estrogens have a key role in prostate development [33,34]. Different studies have related EDC compounds to the development and progression of prostatic disease [35] as well as prostate cancer [36]. In vivo reports have shown that BPA heightens prostate cancer susceptibility in a dose-specific manner [37]. Moreover, epidemiological studies revealed that BPA is positively correlated to prostate cancer risk [24]. Molecular evidence proved that BPA regulates prostate cancer cell proliferation and induces cell cycle arrest in androgen-dependent prostate cancer cell lines (LNCaP, LAPC4) [38], as well as the induction of DNA adducts and promotion of calcium-dependent migration [39] in prostate cancer cells [40]. These effects are highly dependent on the affinity of BPA to the estrogen receptors Erα and Erβ [41] and the androgen receptor (AR). Notably, AR mutations such as the AR protein mutant AR-T877A seem to influence the activity of BPA. At environmentally relevant concentrations (1 nM), this agent is capable of activating the expression of prostate-specific antigen *PSA*, and inducing accelerated tumor growth in vivo, consequently facilitating the biochemical recurrence in tumors after therapy [42,43,44]. BPA proved to possess androgenic and anti-androgenic effects in rodent models [45]. The molecular mechanisms behind this activity remained unclear. To solve this problem Hess-Wilson, et.al. determined the transcriptomic profile of LNCaP cells exposed to BPA 1 nM. This group found that BPA activated mechanisms of cell proliferation by the downregulation of ERβ [46]. Taking into account that ERs are mainly expressed in the male reproductive system, it is worth considering that the effects of BPA on prostate cancer are due to their effects on estrogen receptors [47].

However, this evidence only considers the effects of bisphenol A during the early stages of PCa, where cells are highly sensitive to the action of androgens but fail to describe what happens when androgen-resistant prostate cancer cells are exposed to BPA. It is pertinent to use other models to fully characterize the effect of low-concentration BPA exposure alongside the course of the disease. The PC-3 cell line might be the ideal model to evaluate the action of BPA in advanced stages of prostate cancer. Comparing LNCaP and PC-3 cell lines is a common course of action due to their phenotypic differences since LNCaP cells share characteristics with early-stage prostate cancer (adenocarcinoma), while PC-3 has common features with small-cell neuroendocrine carcinoma (SCNC) [48]. To provide a further understanding of the molecular impact of low-concentration exposure of BPA and its structural analogs BPS and BPF, this work aims to characterize the effect of these exposures on both LNCaP and PC-3 prostate cancer cell lines.

## 2. Results

### 2.1. Low-Concentration Bisphenol Exposure-Induced Cytotoxicity and Produced Changes in the Cell Proliferation Pattern in Androgen-Independent PC-3 Prostate Cancer Cells

To understand the molecular mechanisms of low-concentration (nM) exposure to BPA and analogs through the course of the disease, LNCaP and PC-3 prostate cancer cells were exposed to 1, 5, or 10 nm concentrations of BPA, BPS, and BPF for 48 h. Then, cell viability was measured to determine the cytotoxic effects of BPA and analogs. Androgen-dependent LNCaP prostate cancer cells failed to exhibit statistically significant cytotoxic effects at any exposure (Figure 1A). In contrast, cell viability in PC-3 cells was diminished at 10 nM concentration. It is worth noting that there were substantial differences in the dose-response pattern between both cell lines, as PC-3 cells exhibited a non-monotonic pattern (Figure 1B), and although statistically significant differences were not detected at concentrations of 1 and 5 nM, a similar dose-response pattern was reported in rat prostatic epithelial cells exposed to nM concentrations of BPA [49]. Once the effect over cell viability was determined, 5 nM concentration was chosen for further assessment since it was the highest concentration with no adverse effect observed.

Cell proliferation was measured in prostate cancer cells at 24, 48, 72, and 96 h. LNCaP cells were not affected by any exposure (Figure 1C), and PC-3 cell lines seemed to have a time-dependent differential effect on proliferation; however, a statistically significant difference was only found with the BPA exposure at 24 h (Figure 1D). These results suggest that the effect of bisphenols exposure on prostate cancer cytotoxicity and proliferation might differ depending on the stage of the disease and the concentration of the exposure.

### 2.2. Transcriptomic Effect of Bisphenols Exposure

Total RNA from LNCaP and PC-3 prostate cancer cells exposed to 5 nM bisphenols (BPA, BPS, and BPF) for 48 h was used to characterize the transcriptomic profiles through the microarray technique. After quality assessment and normalization of the microarray data, differential expression analysis was done using empirical Bayes statistics for differential expression (eBayes). Exposure to BPA in LNCaP cells resulted in a total of 2750 differentially expressed genes (DEGs); 1533 were upregulated, and 1217 were downregulated (Figure 2A). Similarly, BPS exposure induced 2416 DEGs, 1922 upregulated and 494 downregulated (Figure 2B), and BPF exposure presented 3931 DEGs, 2847 upregulated and 1084 downregulated (Figure 2C). The expression profile for each exposure was represented in a clustergram, revealing that each compound induced a unique transcriptomic effect in the LNCaP cell line (Figure 2D). The androgen-resistant prostate cancer cell line PC-3 was less sensitive to bisphenol exposure, as fewer DEGs were observed. The exposure to BPA resulted in 738 DEGs, 318 upregulated, and 420 downregulated (Figure 2E). BPS exposure induced the differential expression of 456 DEGs; 208 were upregulated and 248 downregulated (Figure 2F). Finally, in BPF exposure, there were 359 DEGs, 106 upregulated, and 253 downregulated (Figure 2G). The clustergrams illustrating the expression profiles of each bisphenol exposure revealed a unique transcriptomic profile for each bisphenol exposure (Figure 2H).

An upset plot was built to compare the transcriptomic effects of the different exposures to bisphenols in prostate cancer cell lines. The exposure of LNCaP cells to BPS and BPF shared 1004 DE genes, while exposure to three bisphenols shared only 161 DE genes. For PC-3 cell lines, exposures to BPA and BPS shared 60 DE genes, and exposures to BPF and BPS shared 52 DE genes. Only 27 DE genes were shared in the three exposures for the PC-3 cell line. In conclusion, each bisphenol exposure has a differential effect over the transcriptome of prostate cancer cell lines (LNCaP and PC-3), and the possible implications of these effects need to be explored; due to this, we performed the functional characterization of the transcriptomic effect of these bisphenol exposures (Figure 2I).

### 2.3. Pathway Over-Representation Analysis

To further understand the possible role of these transcriptomic changes on prostate cancer cell lines, a pathway over-representation analysis was performed using the EnrichR package [50]. The over-representation analysis determined whether genes from predefined sets, as GO terms, are more present than would be expected in our data. To perform this analysis, the protein-coding DE genes for each bisphenol exposure were split into two gene sets, one containing the overexpressed DE genes (FC > 2) and the other one containing the downregulated DE genes (FC < −2). Once the datasets were split, pathway over-representation was performed, considering that GO terms were overrepresented if they had a *p*-value < 0.05 and were sorted by their combined score.

#### 2.3.1. Upregulated Pathways

In LNCaP cells exposed to BPA, the only statistically overrepresented pathway was activin receptor signaling (Figure 3A) which was previously associated with the regulation of prostatic cell adhesion and viability [51]. BPS disrupted thirteen pathways related to biomolecule metabolism, protein processing, differentiation, and mitosis control (Figure 3B,G). BPF exposure altered sixty-six pathways of biomolecule metabolism, vesicle transport, cell cycle control, vacuole function, and cell response to stimulus (Figure 3C,G). There were only six GO terms shared between BPF and BPS exposure (mainly related to metabolism, vacuolar function, and RNA processing), indicating that the effect of these compounds over the upregulated DE genes was more similar than compared with the other exposures (Figure 3G). In PC-3 cell lines, the main overrepresented pathways due to BPA exposure were epithelial cell-to-cell adhesion (Figure 3D), blood circulation in the case of BPF exposure, and seventeen enriched GO terms (Figure 3F) for the BPS exposure related to biomolecule metabolism (such as nucleotides), immune cell chemotaxis, and protein processing (Figure 3E,G).

#### 2.3.2. Downregulated Pathways

For all the bisphenol exposures, there were more statistically significant enriched pathways than downregulated DE genes used as a query for the EnrichR analysis. In LNCaP cells exposed to BPA, the statistically enriched pathways were related to DNA elongation, metabolism, and repair, as well as response to DNA damage, cell cycle control, mitotic cell cycle phase transition, and mitochondrial function (Figure 4A). For the BPS exposure in LNCaP cells, the main overrepresented pathways were the ones related to sterol and cholesterol import and metabolism, which also disrupted different biomolecule biosynthesis (Figure 4B).

BPF exposure disrupted pathways related to DNA repair, replication, metabolism, cell cycle control, and chromatin organization in LNCaP cells (Figure 4C). In PC-3 cell lines, bisphenol A disrupted pathways related to cell division, proliferation, cell cycle control, DNA damage response, and apoptosis regulation (Figure 4D). Bisphenol S exposure impacted pathways related to apoptotic signaling in mitochondria, phosphate ion transport, protein insertion in mitochondria, and vascular endothelial growth factor (Figure 4E). In the case of BPF exposure, it downregulated genes related to cell cycle control regulation, mitotic phase transition, mitochondrial transport, and DNA damage repair response (Figure 4F). From the upset plot, we determined that there were some common shared enriched pathways among the different bisphenol exposures (Figure 4G). There was one enriched pathway shared between LNCaP exposures to BPA, BPF, and PC-3 exposure to BPS and BPF, which was the GO term GO:0006839, associated with mitochondrial transport. Another shared GO term between four exposures was GO:0030330 (DNA damage response, signal transduction by p53 class mediator) which was shared between PC-3 exposure to BPA and BPF, also between LNCaP exposures to BPA and BPF. There were twenty-three GO terms shared between three bisphenol exposures (Figure 4G), most of them related to cell cycle control, DNA damage response, and biomolecule metabolisms such as GO:0010564 (regulation of cell cycle process), GO:0000086 (G2/M transition of the mitotic cell cycle), GO:0006977 (DNA damage response, signal transduction by p53 class mediator resulting in cell cycle arrest), GO:0007077 (mitotic nuclear membrane disassembly), and GO:0007346 (regulation of mitotic cell cycle), to give some examples (Appendix A). As most of the enriched GO terms shared between three or more bisphenol exposures were related to DNA damage, cell cycle control, and biomolecule metabolism, further assays were performed to elucidate whether the disruption on the transcriptome was capable of inducing a phenotypic effect.

### 2.4. DNA Damage Induction by Exposure to Bisphenols

The bioinformatic analysis confirmed that the exposure to bisphenols A, S, and F at nanomolar concentration was capable of disrupting pathways related to DNA damage repair, elongation, metabolism, and cell cycle control through DNA damage; it was relevant to characterize whether the DNA damage was detectable in cells exposed to these compounds. To answer this question, two different methodologies were used: the Cytokinesis-Blocked Micronuclei Assay (CBMN) and the Comet Assay.

CMBN assay allows the detection of chromosomal breaks or whole chromosome excisions from the nuclear material, as well as bud formation and chromatin bridges. Exposure of 5 nM BPA, BPS, or BPF in LNCaP and PC-3 cells did not affect the micronuclei number (Figure 5A,B). 

As no permanent DNA damage after exposing prostate cancer cells to low concentration of bisphenols was observed by the CMBN assay, we performed an alkaline comet assay to detect repairable DNA insults such as DNA double-strand breaks, DNA single-strand breaks, and incomplete excision repair sites, among others [52]. In this assay, all bisphenols A, S, and F at 5 nM concentration were capable of inducing DNA damage for both cell lines LNCaP (Figure 5C) and PC-3 (Figure 5D), represented by an increase in the moment of the comet tails (relationship between each comet length and signal intensity). All exposures were compared against the vehicle (ethanol), and we included a DNA damage positive control, Mitomycin C (MMC) at 5 µM concentration. These results indicate that the exposure to bisphenols A, S, and F was not capable of inducing irreversible chromosomic DNA damage, but these exposures were capable of increasing the amount of strand DNA damage in prostate cancer cell lines, which confirms the bioinformatic results for the transcriptome analysis.

### 2.5. Cell Cycle Analysis

As results from the pathway over-representation analysis showed that most of the main DE genes were involved in cell cycle control and checkpoint mechanisms, we evaluated the cell cycle in prostate cancer cells exposed to bisphenols. To compare the effects that bisphenol exposure might have on the cell cycle regulation, human lymphocytes were used as G0/G1 phase positive control, and prostate cancer cell lines were treated with the antineoplastic agent nocodazole which inhibits the microtubule synthesis arresting cells in G2/M phase. All groups were compared with negative control (prostate cancer cell lines treated with ethanol) (Figure 6A). Exposition of the androgen-sensitive LNCaP cells to BPA, BPS, or BPF did not affect any cell cycle phase. In contrast, PC-3 cells exposed to 5 nM BPA induced cell cycle arrest in G0/GQ (Figure 6B). Cell cycle inhibition was previously reported in human prostate cancer cells, and the possible mechanism was the downregulation of cyclin D1 expression and the overexpression of cell cycle inhibitors as p21 and p27 [38]. However, further research is still needed to explain this mechanism in both stages; androgen-sensitivity and androgen-independence with more BPA analogs.

## 3. Discussion

In this work, we showed that exposure to bisphenol A, S, and F at nanomolar concentrations has a differential effect as cytotoxic agents in prostate cancer cell lines. PC-3 cells were more sensitive to the cytotoxic effect, presenting a statistically significant reduction in cell viability after bisphenols exposure at 10 nM concentration. In the case of LNCaP cells any bisphenol exposure presented cytotoxic effects. A similar pattern was observed in cell proliferation assays; in LNCaP cells, proliferation was not affected, in contrast to PC-3 cells that presented an increase in cell proliferation after 24 h exposition to BPA (5 nM) and BPF (10 nM). The differences might be explained by specific features of each cell line LNCaP cells share characteristics with early-stage prostate cancer (adenocarcinoma), while PC-3 is a cell line that resembles a small cell neuroendocrine carcinoma (SCNC), including the expression of the androgen receptor (AR) and prostate-specific antigen (PSA) whereas PC-3 cells do not express these proteins [48]. Another substantial difference between cell lines is the presence of hormone receptors besides AR, such as estrogen receptors ER-α (ESR1)and estrogen receptors ER-β (ESR2). In vitro models demonstrated that LNCaP cells express ER-β but not ER-α, while PC-3 cells express ER-β at the mRNA level [53]; however, at the protein level, LNCaP cells failed to express any of the estrogen receptors but express AR. PC-3 cells express ER-α but not AR [54]. It was previously reported that ER-α tends to promote prostate tumorigenicity [55]. Interestingly, in other cellular models (HeLa), it was proved that bisphenol A has dual effects as an agonist and antagonist for ER-α, activating the receptor activity at low concentrations [56]. This evidence supports that PC-3 is more sensitive to both cytotoxic and proliferative effects than LNCaP cells which is consistent with our observations. Exposure to bisphenols at 5 nM did not exhibit any cytotoxic effects; therefore, the remaining analysis was performed by exposing PCa cells to bisphenols at this concentration. We characterized the effect of bisphenols on transcription on LNCaP and PC-3 since previous evidence showed that the transcriptional profile of cell cycle regulation and proliferation could be impacted by BPA [46], as an increase in cell proliferation but not cell proliferation in LNCaP cells. This difference can be explained due to the time difference between these studies, as they exposed LNCaP cells to bisphenols for 24 h, and we performed most of the assays after 48 h exposure, considering the time to complete one cell replication cycle.

Microarray expression analyses demonstrated that each bisphenol exposure has a unique transcriptional profile. LNCaP cells had a higher rate of DEGs compared to PC-3 cells, which might be explained by the inherent characteristics of both cell lines and their expression of hormone receptors. It has been reported, in other models, that BPA has estrogenic activity but may also act as an anti-androgenic compound [57]. Additionally, it was reported that BPA, BPS, and BPF induce ER-α/ERβ, antagonizing AR activity [58]. As LNCaP cells present estrogenic and androgenic functions, more differentially expressed genes were found when exposed to bisphenols, in contrast to PC-3 cells which only express ER-α, and therefore fewer DEGs were observed after bisphenol exposure.

To determine the potential pathways of the DEGs, we performed a pathway over-representation analysis and found that there were less statistically significant enriched GO terms when using the upregulated DEGs as query terms compared to the downregulated ones. The LNCaP cells exposed to BPF had more upregulated GO terms than all the other exposures, and most of the statistically significant enriched pathways were related to biomolecule metabolism. There were no clear shared patterns of pathway enrichment among the bisphenol exposures for the upregulated DEGs. In contrast, the downregulated DEGs induced the dysregulation of more GO terms. In the case of downregulated pathways, there were more biological functions shared among all the exposures, and most of them were related to cell cycle regulation, DNA damage repair mechanisms, DNA elongation and metabolism, and biomolecule metabolism. It was previously reported that bisphenols are capable of inducing DNA damage in different cell models through oxidative stress and disrupting cMYC and CTNNB1 signaling pathways, inducing cell cycle arrest [59,60,61,62]. In normal epithelial cells, evidence proved that BPA induces DNA damage (RWPE-1) [63] and cell cycle arrest in human non-transformed epithelial prostate EPN cells and prostate cancer cells (LNCaP) [38]. However, there is a lack of evidence regarding the effect of these three structural analogs (BPA, BPS, and BPF) in both androgen-sensitive and independent prostate cancer cells LNCaP and PC-3.

Our observations show the effect of bisphenols exposure on DNA damage and cell cycle control. Despite the fact that permanent chromosomic DNA excisions or damage in the micronuclei assay were not apparent, there was a tendency towards an increased amount of micronuclei after bisphenol exposure, we found a statistically significant increase of DNA breaks, as shown by the comet assay for all bisphenol exposures suggesting bisphenols might affect DNA strand breaks both, single or double, or they may induce an incomplete excision repair sites. The accumulation of small DNA breaks might be able to induce larger chromosomal abnormalities, reflected in the increasing amount of micronuclei number; this latter was only observed as a trend. The effects of bisphenols on DNA damage were consistent with those from the transcriptomic analyses, where we found a downregulation in DNA damage repair genes. Similar results were reported in normal epithelial prostate cells RWPE-1 exposed to BPA, BPS, and BPF where repair proteins, OGG1, Ape-1, MyH, and p53 involved in the base excision repair were downregulated during bisphenol exposures [63]. Similarly, Chen Yin-Kai et al. reported downregulation of TP53 and CDKN1A, putatively promoting DNA damage [59].

LNCaP cells showed greater DNA damage than PC-3 cells, as shown by the comet assay.. In terms of toxicological potency, the three bisphenols did not show a statistically significant difference in the comet tail moment; however, there is a tendency where bisphenol F exposure induced slightly higher comet tails in both prostate cancer cell lines. In the case of PC-3 cells, cell cycle analysis revealed an induction of cell cycle arrest in the G0/G1 stage. For the other treatments, there was no statistically significant difference. Integrating the pathway over-representation analysis and the cell cycle results for PC-3 cells exposed to BPA, we identified the downregulation of important genes for cell cycle control, such as TP63, MYO19, and CDC25 were downregulated after the exposure. In fact, TP63 ablation has been related to G1 cell cycle arrest [64]. With these assays, we confirmed that there was single and double-strand DNA damage induction in both LNCaP and PC-3 cells and that the cell cycle was only affected by BPA in the androgen-independent prostate cancer cell line PC-3. Cell cytotoxicity, cell proliferation, transcriptomic profiles, bioinformatic pathway analysis, DNA damage, and cell cycle assays indicate that exposure to low-concentration bisphenols might have a differential effect in prostate cancer cells, which may correlate in vivo with a distinctive disease stage. This can have an explanation considering the androgenic, anti-androgenic, and estrogenic activities, other endocrine disrupting mechanisms, and epigenetic changes of bisphenols differentially impacting PC-3 and LNCaP [65,66].

Further research, such as in vivo models and mechanistic molecular tests, will be needed to fully understand the mechanistic differences between these exposures. Nevertheless, this work provides a first insight highlighting the impact of bisphenol at the transcription level for prostate cancer progression. Environmental toxicology, omic sciences, and translational medicine should be integrated to provide more consistent evidence related to EDCs and their impact on non-communicable diseases.

## 4. Materials and Methods

In this study, prostate cancer cell lines LNCaP and PC-3 were exposed to environmentally relevant concentrations of BPA, BPS, and BPF (1, 5, and 10 nM) to determine the exposure effect over cytotoxicity, proliferation, transcriptome, DNA damage, and cell cycle.

### 4.1. Chemicals

Bisphenol A (4,4′-Isopropylidenediphenol; Cas. 239658-50G; Lot #MKBX9458V), Bisphenol S (4,4′-Sulfonyldiphenol; Cas. 103039-100G; Lot # MKCF0795) and Bisphenol F (4,4′-Methylenediphenol; Cas. B47006-1G; Lot #00125KJ) were acquired from Sigma–Aldrich company [67,68,69]. To treat cells with these compounds, bisphenols were diluted in absolute ethanol as a vehicle.

### 4.2. Cell Culture

Prostate cancer cell lines LNCaP (CRL-1740) and PC-3 (CRL-1435) were acquired from the American Type Culture Collection (ATCC, Inc., Manassas, VA 20110-2209 USA) [70,71]. All cells were cultured with RPMI 1640 media (pH = 7.4) (Sigma–Aldrich, Inc., St. Louis, MI, USA, R8758) [72] with 10% of fetal bovine serum (FBS) and incubated under standard conditions (5% CO_2_, 37 °C).

### 4.3. Cell Viability Assay

Approximately 1 × 10^4^ and 5 × 10^3^ prostate cancer cells LNCaP and PC-3 were seeded into 96-well plates. Twenty-four hours after seeding, cells were hormone depleted using RPMI 1640 media without phenol red and FBS charcoal stripped (1%). A low concentration of FBS is crucial to perform this assay to starve cells avoiding cell proliferation. After hormone depletion and starvation, cells were treated with bisphenols added to their cell culture media to reach final concentrations of 1 nM, 5 nM, and 10 nM (vehicle final concentration was 1% *v*/*v*).

Forty-eight hours after treatment, cell culture media was removed, and cells were washed with phosphate-buffered saline (PBS) 1X, then fresh culture media prepared with MTT (3-(4,5-dimethyl thiazol-2-yl)-2,5-diphenyl tetrazolium bromide (Abcam, Inc., Cambridge, MA, USA, ab211091) reactive [73] was added. For each well, an equivalent of 15 µL of MTT reactive per well was added, and they were diluted in 35 µL of cell culture fresh media. After this, cells were incubated for two hours under standard conditions (5% CO_2_, 37 °C) until formazan crystals (purple) were observed. Then, 100 µL of MTT reactive solvent was added to each well of the plate. Cells were incubated at room temperature until the cells had been lysed and the purple crystals had been dissolved. The absorbance was measured at 570 nm using a microplate reader. Cell viability was normalized against the cell viability of cells without any treatment.

### 4.4. Cell Proliferation

Approximately 2.5 × 10^3^ prostate cancer cells LNCaP and PC-3 were seeded into 96-well plates. Twenty-four hours after seeding, cells were hormone depleted (but with 10% of charcoal-stripped FBS) and treated with bisphenols A, S, and F at 1 nM, 5 nM, and 10 nM. The cell proliferation assay using crystal violet was performed at four different times for each cell line (24, 48, 72, and 96 h), following the next protocol. After each one of the established times for treatments, the cell culture media was removed. Then, cells were washed twice with 50 µL of PBS 1× After washing, cells were fixed with 50 µL of 4% paraformaldehyde for ten minutes. Once fixed, we removed paraformaldehyde and let cells dry at room temperature. Then we washed cells again with 50 µL of PBS 1×. PBS was removed, and cells were stained with 0.02% crystal violet in deionized water (50 µL/well) for 10 min. The excess was discarded by washing cells again with PBS 1×. The cell-bound dye was redissolved in 33% *v*/*v* acetic acid/water. In the end, optical density was measured at λ = 595 nm. Cell proliferation was normalized against the cell viability of cells without any treatment.

### 4.5. RNA Extraction and Quality Assessment

RNA purification was performed using TRI reagent (Sigma–Aldrich, Inc., St. Louis, MI, USA) [74], following the manufacturer’s protocol for cells cultured in monolayer. Once extracted, the RNA concentration and purity were measured using a NanoDrop spectrophotometer (Thermo Fisher Scientific, Inc., Waltham, MA, USA) [75], determining the sample absorbance at a wavelength of 260 nm, and for purity evaluation, we measured the absorbance ratio at 260/A280 nm and 260/A230 nm. We considered that a sample was relatively pure when the 260/A280 nm ratio was ~2.0, and the 260/230 ratio was between 2.0 and 2.2. Then the RNA quality was determined by automated electrophoresis, using the Agilent, Inc., Santa Clara, CA, USA, 2100 Bioanalyzer Instrument [76], following the manufacturer’s indications. Samples with an RNA integrity number (RIN) ≥ 9.0 were considered suitable for the microarray assay.

### 4.6. Microarray Assay

To determine the exposure effect of bisphenols A, F, and S on the transcriptome of prostate cancer cell lines, we used the Clariom™ D Assay, human from (Thermo Fisher Scientific, Inc., Waltham, MA, USA), which contains probes to recognize >542,500 human transcripts, providing a wide view of human transcriptome expression profile [77]. We followed the manufacturer’s protocol for sample preprocessing, labeling, hybridization, and microarray imaging [78].

### 4.7. Data Analysis

#### 4.7.1. Differential Expression Analysis

Data were analyzed using Thermo Fisher Scientific’s Transcriptome Analysis Console (TAC 4.0.2). Raw data were processed for normalization and gene-level analysis using the robust multi-array average (RMA) included in the TAC 4.0.2 software. Each experimental condition has three independent microarray replicates. Differentially expressed transcripts were calculated using the means of the gene expression signals to obtain a fold change (FC) of the expression values. Statistical analysis was done using the eBayes limma method included in the TAC 4.0.2 software. Those FC values ≥ 2, ≤−2, and with a gene-level *p*-value ≤ 0.05 were considered significant. Volcano plots and clustergrams (heat maps) were obtained from the TAC 4.0.2 software [79].

#### 4.7.2. Pathway Over-Representation Analysis

The DEGs from each bisphenol exposure were filtered into two new data sets, one containing the upregulated DEGs and the other with the downregulated DEGs. Each new dataset was used as input for the EnrichR package in R [50] to perform the pathway over-representation. The GO_biological_Process_2018 library was considered to perform the over-representation analysis. Pathways were considered statistically enriched if they had a *p*-value < 0.05 and were sorted considering their odd ratio. Dot plot visualization was built using the ggplot2 R package [80].

#### 4.7.3. Cytokinesis-Blocked Micronuclei (CBMN) Assay

The CBMN assay is a worldwide validated method used to analyze genotoxic effects. The assay was conducted according to the conditions described by Fenech M. et al. [81] but adapted to monolayer-grown cells. The 0.5 × 10^6^ cells were seeded onto a cover slip in 60 mm dishes. Cells were exposed to bisphenols A, F, and S (1 nm, 5 nM, and 10 nM) for 48 h. After the treatment time, cytochalasin B (3 mg/mL final concentration) was added for 24 h. Cells on the coverslips were fixed in methanol/acetic acid (3:1). The cover glasses were stained with Wright’s colorant and were mounted on slides for microscopy evaluation. Cell proliferation kinetics were analyzed by determining the frequency of mononucleated (Mono), binucleated (Bi), and polynucleated (Poly) cells. The cytostatic activity was determined by calculating the nuclear proliferation index (NPI) with the formula NDI = [M + 2(BN) + 3(P)]/N, with N being the total counted cells. The frequency of MN was determined in 500 binucleated cells per treatment group, and this measure is associated with DNA damage.

#### 4.7.4. Alkali Comet Assay

To perform an alkali comet assay, we followed the indications described by Lu, Y. et al. [82], adapted to monolayer-grown prostate cancer cell lines LNCaP and PC-3. Cells were exposed to bisphenol A, S, and F (1, 5, and 10 nM) for 48 h. After exposure, the culture was detached from the monolayer and resuspended in 1 mL of RPMI 1640. Then, they were incubated for one hour with 10 µL of N- hydroxyurea. After incubation, 150 µL of low melt-point point agarose (LMP) was added to each sample. A total of 75 µL of the solution was added to previously prepared low melting point agarose in glass slices. Glass slices were covered with a coverslip and incubated at 4 °C for 5 min. After incubation, the other 75 µL were added to the glass slices and incubated for another 5 min. Glass slices were submerged into 50 mL of lysis solution (2.5 M NaCl, 100 mM Na2EDTA, 100 mM Tris) with 5 mL of dimethylsulfoxide (DMSO) and 0.5 mL of Triton X-100 solution. Glass slices were incubated in this solution for an hour. Once the cells were mounted into the slices with agarose, electrophoresis was performed for 40 min at 25 V. After electrophoresis, glass slices were washed with 2 mL of Tris pH 7.4 and were fixed with ethanol. Slices were dried under environmental conditions.

To read the samples, 10 µL of bromide ethidium was added to each sample to stain the nuclei chromatin and observe the DNA fragments that compose the comet. We determined the comet presence and moment (relationship between length and intensity) with the Perceptive Instruments Comet Assay IV image analysis systems [83].

#### 4.7.5. Flow Cytometry and Cell Cycle Analysis

Cell cycle analysis was performed through the propidium iodine method coupled to flow cytometry. Cells were exposed to bisphenol A, S, and F (1, 5, and 10 nM) for 24 and 48 h. Then, they were digested with 1 mL of trypsin and centrifuged at 1200 rpm for 8 min at 4 °C. The cell pellet was resuspended in 40 µL of ice-cold PBS, and then 460 µL of cold ethanol was added. The suspension was stored at −20 °C for at least 24 h. After the storage time, cells were centrifuged at 2000 revolutions per minute (rpm) for 6 min at 4 °C. Then cell pellets were resuspended into a staining solution (propidium iodine 2 mg/mL) and incubated in dark conditions for 30 min. Each sample was loaded into the cytometer, and results were analyzed with Verity’s ModFit LT software version 3.3 [84].

## 5. Conclusions

Bisphenol A structural analogs BPF and BPS are widely known to exert biological effects; therefore, it is imperative for the scientific community to describe the consequences of their exposure and to fully understand the molecular mechanisms that might relate the bisphenols exposure to disease development. This research provides an initial insight into the effects of exposure to low-concentration bisphenol A structural analogs in two different stages of an in vitro model of prostate cancer. Further molecular research will be needed to fully assess whether BPA structural analogs might have a differential effect in prostate cancer, depending on the stage of the disease.

## Figures and Tables

**Figure 1 ijms-24-09462-f001:**
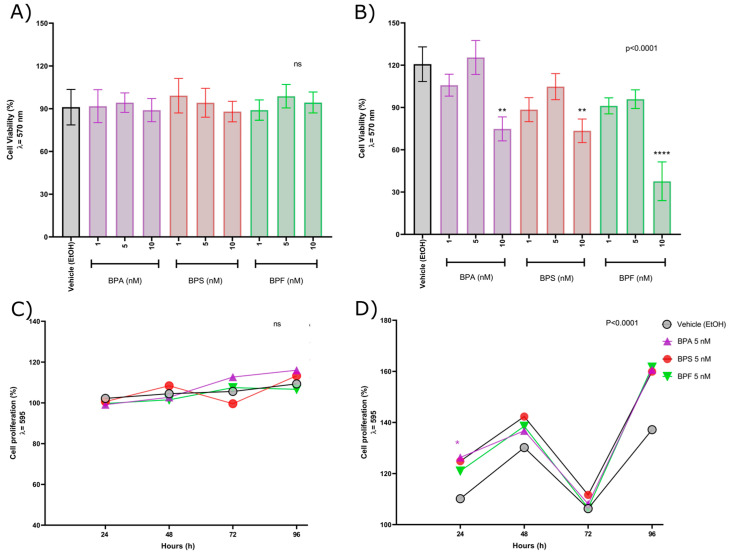
Cell viability in prostate cancer cell lines: (**A**) Cell viability in the androgen-sensitive prostate cancer cell line LNCaP; (**B**) Cell viability in the androgen-independent prostate cancer cell line PC-3. In both cases, cells were exposed for 48 h to each bisphenol (A, S, and F) at 1, 5, and 10 nM concentrations. (**C**) Cell proliferation in the androgen-sensitive prostate cancer cell line LNCaP; (**D**) Cell proliferation in the androgen-independent prostate cancer cell line PC-3. In both cases, cells were exposed for 24, 48, 72, and 96 h to each bisphenol (A, S, and F) at 1, 5, and 10 nM concentrations. An ordinary one-way ANOVA (cell viability) and an ordinary two-way ANOVA test (cell proliferation), followed by Dunnett’s multiple comparison tests, were performed to determine the differences between each group versus the control group, considering an α = 0.05 and *p* < 0.05. The amount of asterics indicate how high is the significance. ** *p* ≤ 0.01, **** *p* ≤ 0.0001.

**Figure 2 ijms-24-09462-f002:**
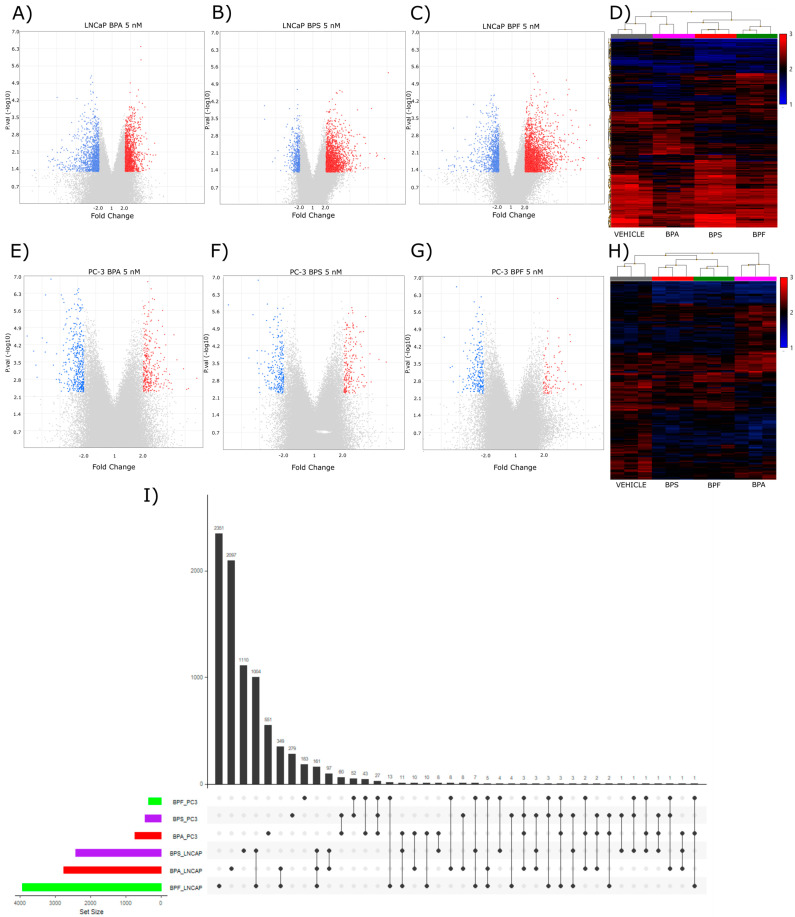
Differential expression analysis of prostate cancer cell lines (LNCaP and PC-3) exposed to bisphenols A, S, and F. Volcano plots present the differentially expressed genes upon the bisphenol exposure, blue dots represent downregulated genes and red dots represent upregulated genes. Heatmaps reproduce expression profiles for differentially expressed genes when comparing the control (vehicle) against each bisphenol exposure. Volcano plots for LNCaP cells exposed to (**A**) BPA 5 nM, (**B**) BPS 5 nM, (**C**) BPF 5 nM, and (**D**) expression profiles for LNCaP cells exposed to bisphenols A, S, and F. Volcano plot for PC-3 cells exposed to (**E**) BPA 5 nM, (**F**) BPS 5 nM, (**G**) BPF 5 nM, (**H**) Expression profiles for PC-3 cells exposed to bisphenols A, S, and F. Upset plot showing the shared DE genes between all bisphenol exposures. Dots indicate the levels of interactions, and bars indicate the number of DE genes (**I**).

**Figure 3 ijms-24-09462-f003:**
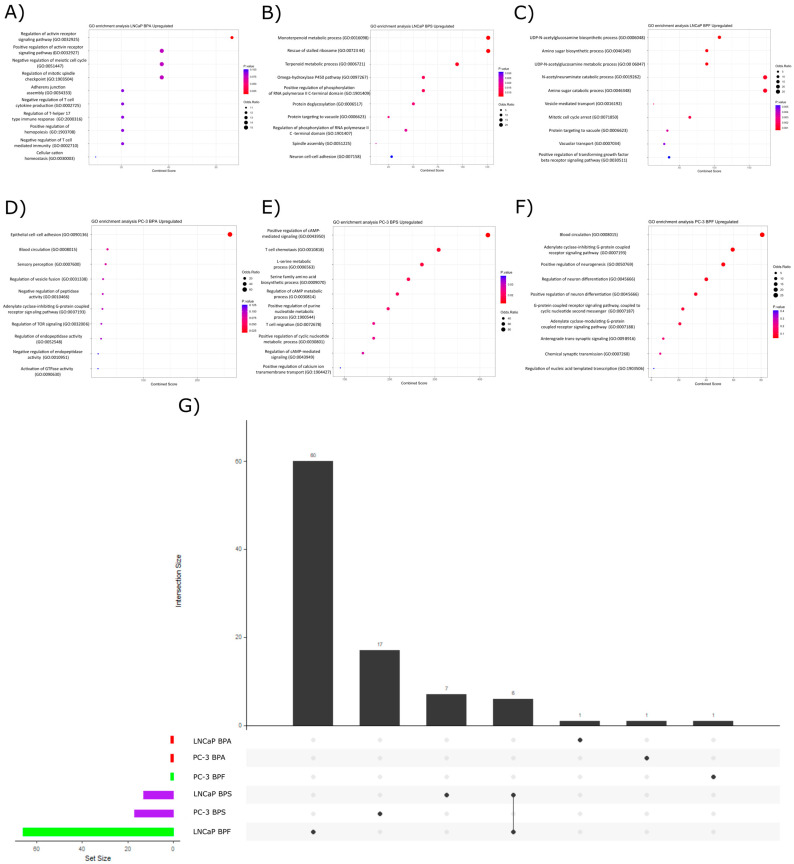
Pathway over-representation analysis from upregulated DE genes. Dot plots represent the enriched pathways. The size of each dot indicates the odd ratio, and the color represents the *p*-value for each pathway. Dot plots for LNCaP cells exposed to BPA (**A**), BPS (**B**), and BPF (**C**) indicate the top twenty enriched pathways for each exposure. For the PC-3 cell line, the main overrepresented pathways were represented by the dot plots (**A**) for BPA (**E**), BPS (**F**), and BPF (**G**) exposure. The upset plot with the main overrepresented pathways for each bisphenol exposure is represented in Figure (**G**).

**Figure 4 ijms-24-09462-f004:**
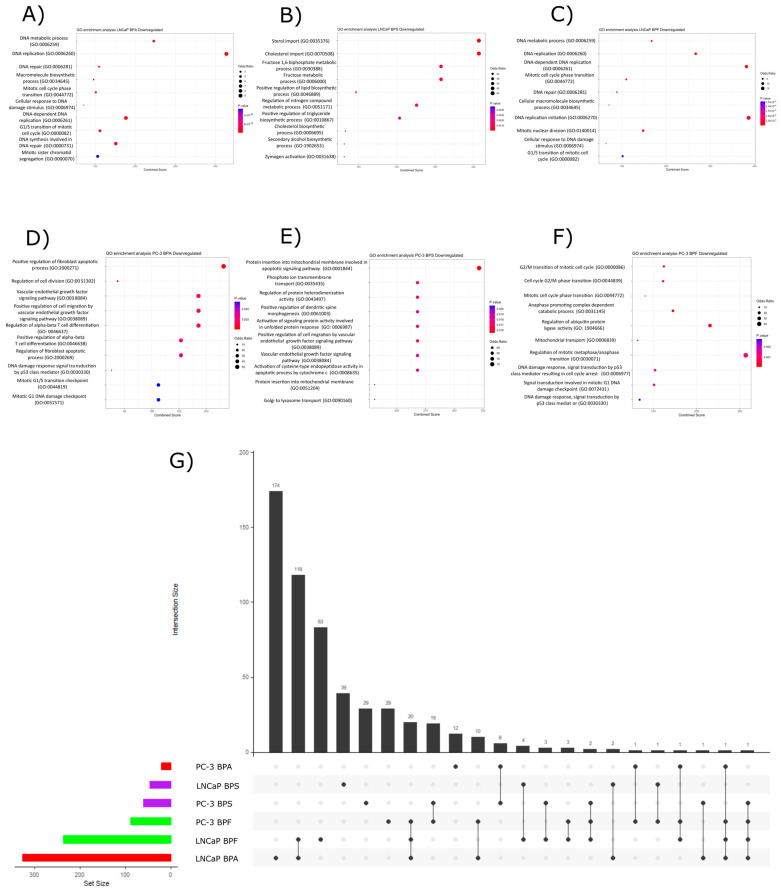
Pathway over-representation analysis from downregulated DE genes. Dot plots for LNCaP cells exposed to BPA (**A**), BPS (**B**), and BPF (**C**). In these figures, the top twenty enriched pathways for each exposure were represented. For the PC-3 cell line, the main over-represented pathways were represented by the dot plots for BPA (**A**), BPS (**E**), and BPF (**F**) exposure. The upset plot with the main overrepresented pathways for each bisphenol exposure is represented in Figure (**G**).

**Figure 5 ijms-24-09462-f005:**
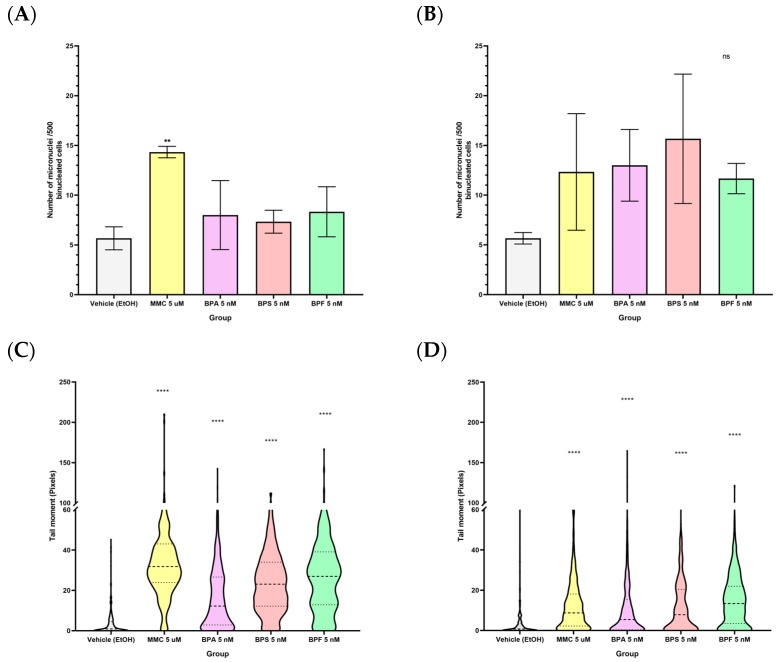
DNA damage analysis in prostate cancer cell lines. Number of micronuclei per five hundred bi-nucleated cells for both prostate cancer cell lines LNCaP (**A**) and PC-3 (**B**). Moment of the comet tails for comet assay analysis in LNCaP (**C**) and PC-3 (**D**) cells. (MMC 5 μM = mitomycin 5 micro Molar). The amount of asterics indicate how high is the significance. ** *p* ≤ 0.01, **** *p* ≤ 0.0001.

**Figure 6 ijms-24-09462-f006:**
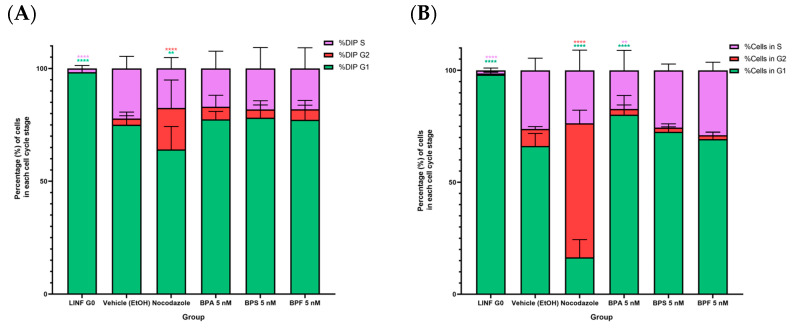
Cell cycle analysis for prostate cancer cell lines LNCaP (**A**) and PC-3 (**B**) exposed to bisphenols A, S, and F at 5 nM. Bisphenol A was capable of inducing cell cycle arrest in PC-3 cells. * The amount of asterics indicate how high is the significance. ** *p* ≤ 0.01, **** *p* ≤ 0.0001.

## Data Availability

The data presented in this study are openly available in the Gene Expression Omnibus (GEO) database, reference number GSE233165.

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
