# Peer review of "Transcriptome-Wide Analysis of Low-Concentration Exposure to Bisphenol A, S, and F in Prostate Cancer Cells"

_ijms, 2023, doi:10.3390/ijms24119462_

Round 1
Reviewer 1 Report
“Transcriptome-wide analysis of low-concentration exposure to Bisphenol A, S and F in prostate cancer cells” by Cortes-Ramirez et al. examined the effects of low dose bisphenol analogues BPA, BPS and BPF in androgen-sensitive LNCaP and androgen-insensitive PC3 prostate cancer cell lines. While the toxicological effects of bisphenol A has been well studied, it is still interesting to see the effects of low doses of other analogues in cancer cells. After reviewing the manuscript, I do have concerns about the geneset analyses regarding some of the DEGs. The PC3 datasets all seem to give genesets with very few genes. Some appears to be only one gene. Perhaps the number of genes for input into the GO analyses is insufficient. Reanalysis with a larger set (with broader cutoff) could help. Otherwise, I think they should be shifted to the supplementary section. The DNA damage section was quite interesting. It is quite alarming that even with such a low dose, there appears to be a significant number of DNA breaks in the comet assay with the bisphenol treatments. However, the discussion didn’t really go into much depth regarding this. There should more discussion on DNA damage aspects and comparison with other studies who have also looked into DNA damage. The discussion is also a bit lacking as the discussion on the differences between LNCap and PC3 was quite brief.
Overall, I think the geneset analysis needs to be reexamined and some of the discussion needs to be expanded to give a more coherent story as I am unable to grasp the message of this paper. Some of the images (Figure 5 and 6) are way too blurry and need to be fixed. Fonts on the figures are also way too small. Please refer to the points below for more details. I would suggest that this manuscript go through major revisions before publication.
Specific points:
Line 38 – suggest “it is” to “which is”
Line 52 – suggest change “might be” to “is” since BPA is well known to be linked to human diseases for many years
Line 64 – “Thus, it is not surprising…”
Line 75 – Why “in synthesis”? Do you mean “As a result”?
Section 2.1 – The three doses of 1, 5, 10 nM are insufficient to generate a dose response, so the selection of 5nM seems quite arbitrary.
Line 10
2-108 – suggest splitting this long sentence
Line 131 – How was the 48 hour timepoint chosen?
Figure 1b – How can the cells have higher than 100% viability? What is it normalized to?
Section 2.2 – PC3 showed more cytotoxicity than LNCaP but showed less DEGs. That is interesting.
Line 191 – missing comma
Figure 2-4 – font is way too small to see, and the figures appear to be quite blurry. Please make sure the images are high enough resolution.
Line 251 – missing comma
Figure 5 – What is MMC? Mitomycin C? Please clarify.
Section 2.3 (1st point) – When examining the GO term analysis from the geneset, it appears that many of the high rank genesets are only driven by a few genes. Some of the PC3 datasets listed only one gene in the output ranks, which suggest that there is not enough input at all. How many genes are used as input for these analyses? If you looked at the adjusted p-value for BPA and BPS for PC3, it’s at 0.2 which is basically noise. The only meaningful datasets are the BPF datasets and BPA_lncap.
Section 2.3 (2nd point) - If there are not enough input genes, I suggest lowering the fold change cut off from 2 to 1.5 (or 1.75). Can you confirm that the fold change value of 2 is linear, not log2 scale? That would make a huge difference in terms of the actual fold change.
Figure 5 and 6 – please clarify what the asterisk means in the figure legend. Are they statistically significant against vehicle control?
Section 2.5 – The results here do not seem too significant. I suggest putting them to supplementary section.
Discussion – PC3 was more sensitive to the cytotoxic effects of the bisphenol but showed lower DEGs levels. Why is that so? Is that explained by the presence of ERalpha or ERbeta?
Line 370 – Just because it is not significant doesn’t mean it is not there. It definitely looks like there is a trend in micronuclei increase (if you increase the sample size, it is likely to reach significance!). These are showing much bigger DNA damage (gross chromosomal instability) than comet tail assay (which includes smaller breaks) so they obviously will have more variability. Since comet tail assay showed significant difference and micronuclei showed strong trend, it is safe to say there are damages in both larger (chromosomal) and smaller scales (local DNA breaks). Accumulations of small DNA breaks can cause large chromosomal abnormalities as well.
Discussion – If DNA damage was indeed detected, why would there be downregulation of DNA repair genes? It would make sense that the cells would upregulate the repair genes instead to counter against DNA damage. An alternative explanation is that the DNA repair genes may be downregulated causing more baseline level of DNA damage (which is occurring all the time but at higher frequencies with bisphenol treatment). Is there any published work to support the second explanation?
Overall, it is mostly ok. Some editing on the English is needed.
Author Response
Manuscript ID: ijms-2350336
Dear Editor and Reviewers,
We really appreciate the time all of you spent reviewing our work as well as all the suggestions made to the manuscript. We gladly accepted most of the reviewer´s comments and a point-by-point report is provided. An appropriate reply to each comment is given in detail. We strongly believe that these modifications have improved the manuscript. A new version of the manuscript with the corrections was elaborated.
1st Reviewer specific comments:
Line 38 – suggest “it is” to “which is”.
This suggestion is appropriate, and the change was made in line 38. “Which is also used as plasticizer for thermoplastic polymers.”
Line 52 – suggest change “might be” to “is” since BPA is well known to be linked to human diseases for many years.
This change was done in line 52. “BPA exposure is involved in different human diseases…”
Line 64 – “Thus, it is not surprising…”
This change was accepted in line 64.
Line 75 – Why “in synthesis”? Do you mean “As a result”?
The change was made in line 75. “In synthesis” was replaced by “As a result.”
Section 2.1 – The three doses of 1, 5, 10 nM are insufficient to generate a dose response, so the selection of 5nM seems quite arbitrary.
While it is true that our study utilized only 0,1,5 and 10 nM concentrations which are not enough to generate a dose response curve, these doses represent environmentally relevant concentrations of the bisphenols according to the occupational studies that we cited in the introduction. We acknowledge that the lack of a complete dose response curve is a limitation for our study, however we believe that our findings still provide valuable insights into the toxic effects of the toxin at these concentrations.
Line 102-108 – suggest splitting this long sentence.
We considered this suggestion was so valuable, therefore the sentence was splitted into
“To solve this problem Hess-Wilson, et.al., determined the transcriptomic profile of LNCaP cells exposed to BPA 1nM. This group found that BPA activated mechanisms of cell pro-liferation by the downregulation of ERβ “
Line 131 – How was the 48-hour timepoint chosen?
In our study, we chose the 48-hour time point based on our preliminary experiments. Previously we performed cell viability assays to determine the amount of time our cells complete a full cycle (doubled their numbers), and we determined that in a period of 48 our cells doubled their numbers. For cytotoxicity assay we decided to block cell proliferation by using cell starvation (reducing FBS to 1%), however we decided to be consistent with this time for all the assays.
Figure 1b – How can the cells have higher than 100% viability? What is it normalized to?
We appreciate this observation. Cell viability was normalized to cells without any treatment, not even the vehicle (ethanol). However, this description was also included in the methodology description (lines 455 & 470).
Section 2.2 – PC3 showed more cytotoxicity than LNCaP but showed less DEGs. That is interesting.
As PC-3 cells only express one estrogen receptor and no androgen receptor, they are less sensitive to the genomic effects of endocrine disruption. However, they are also affected by the rapid non genomic effects of endocrine disruptors.
Line 191 – missing comma
We added the comma at the beginning of the paragraph and changed the expression “in synthesis” to “in conclusion.” (Line 195)
Figure 2-4 – font is way too small to see, and the figures appear to be quite blurry. Please make sure the images are high enough resolution.
Thanks for the comments all the figures 2-4 were made again, increasing the fonts and correcting the blurry. Also, we showed less significant enriched pathways.
Line 251 – missing comma
We added the missing comma in line 255.
Figure 5 – What is MMC? Mitomycin C? Please clarify.
“All the exposures were compared against the vehicle (Ethanol) and we included a DNA damage positive control, Mitomycin C (MMC) at 5 µM concentration”. Was included in line 286.
Section 2.3 (1st point) – When examining the GO term analysis from the geneset, it appears that many of the high rank genesets are only driven by a few genes. Some of the PC3 datasets listed only one gene in the output ranks, which suggests that there is not enough input at all. How many genes are used as input for these analyses? If you looked at the adjusted p-value for BPA and BPS for PC3, it’s at 0.2 which is basically noise. The only meaningful datasets are the BPF datasets and BPA_lncap.
Thank you for the comments regarding the GO term analysis in our study. For building the analysis we used only the protein DE genes to find out the main protein coding genes related to the phenotype. There were hundreds of DE genes used as input for the analysis in the case of LNCaP cells, and tens of DE genes in the case of PC-3. We understand that according to the adjusted p-value there were few DE datasets. For this reason, we conducted a gene set enrichment analysis. This kind of analysis uses the raw signals of each microarray probe. From this analysis we obtained hundreds of DE gene sets in the case of LNCaP cells, and tens of DE gene sets in the case of PC-3 cells. However, when analyzing the most repeated gene sets, they were also related to DNA damage repair and cell cycle regulation. We can include these results as a supplementary analysis because there were too many enriched GO terms when including the non-protein coding transcripts.
We include the enriched GO terms as supplementary in this response.
Section 2.3 (2nd point) - If there are not enough input genes, I suggest lowering the fold change cut off from 2 to 1.5 (or 1.75). Can you confirm that the fold change value of 2 is linear, not log2 scale? That would make a huge difference in terms of the actual fold change.
We really appreciate this observation, previously we performed the DE analysis considering a log fold change of 1.5. However, the pathway enrichment analysis results were similar to the results at FC of 2.0. We found that, as well as the reported analysis, when we considered a FC of 1.5 the high rank gene sets were only driven by few genes. For this reason, we decided to keep the analysis with the FC of 2.0 as it improved the statistical astringency.
Figure 5 and 6 – please clarify what the asterisk means in the figure legend. Are they statistically significant against vehicle control?
This comment was appreciated and implemented in the article (lines 293 and 312).
Section 2.5 – The results here do not seem too significant. I suggest putting them in the supplementary section.
The results for PC-3 cells exposed to BPA were significantly relevant. Therefore, we decided to keep this graph as part of the article.
Discussion – PC3 was more sensitive to the cytotoxic effects of the bisphenol but showed lower DEGs levels. Why is that so? Is that explained by the presence of ERalpha or ERbeta?
We conclude that the lower DEGs levels were associated with the lack of ER beta and AR hormone receptors. As well as BPA, exhibited dual effects as agonist and antagonist of the proliferative ERalpha receptor in other cellular models (Lines 326-335).
Line 370 – Just because it is not significant doesn’t mean it is not there. It definitely looks like there is a trend in micronuclei increase (if you increase the sample size, it is likely to reach significance!). These are showing much bigger DNA damage (gross chromosomal instability) than comet tail assay (which includes smaller breaks) so they obviously will have more variability. Since comet tail assay showed significant difference and micronuclei showed strong trend, it is safe to say there are damages in both larger (chromosomal) and smaller scales (local DNA breaks). Accumulations of small DNA breaks can cause large chromosomal abnormalities as well.
We really appreciate your comments on this matter. We improved the discussion as you suggested:
“Although we did not find statistically significant permanent chromosomic DNA excisions or damage in the micronuclei assay, there was an increasing tendency in the amount of micronuclei for both cell lines. In the case of DNA breaks, we found a statistically significant increase in the moment of comet tails for all exposures. These results indicate that bisphenols might have an effect on DNA strand breaks (single and double) or they induce incomplete excision repair sites. The accumulation of small DNA breaks might be able to induce larger chromosomal abnormalities, reflected in the increasing amount of micronu-clei number, even though it is just a trend”. (LINES 374-381)
We consider that the statistical significance, might not be reached by increasing the sample size as we performed the assay with three technical replicates and three biological replicates, and we counted each sample twice in a double-blind study.
Discussion – If DNA damage was indeed detected, why would there be downregulation of DNA repair genes? It would make sense that the cells would upregulate the repair genes instead to counter against DNA damage. An alternative explanation is that the DNA repair genes may be downregulated causing more baseline level of DNA damage (which is occurring all the time but at higher frequencies with bisphenol treatment). Is there any published work to support the second explanation?
Thank you for your kind comments, as you suggested we improved this section in the discussion part of the manuscript:
“The effects on DNA damage are consistent with the results from the transcriptomic analysis where we found downregulation in the DNA damage repair genes. Similar results were reported in normal epithelial prostate cells RWPE-1 exposed to BPA, BPS and BPF. In this model DNA repair proteins (OGG1, Ape-1, MyH & p53) involved in the base excision repair pathway were down-regulated in all of the bisphenol exposures [61], also Chen Yin-Kai et al., reported a downregulation in the TP53 and CDKN1A promoting DNA damage [57].”
(Lines 381-388)
Reviewer 2 Report
This manuscript describes a very interesting study dealing with transcriptome-level analysis of low-concentration exposure to 2 bisphenol A, S, and F in prostate cancer cells. While a literature review is needed to update the references, here are some references that should be included in the references:
- DOI: 10.1186/s12263-017-0555-5
DOI: 10.1177/1559325815610582
DOI: 10.3390/ijms24065342
DOI: 10.3390/ijms23031216
DOI: 10.1016/j.reprotox.2010.03.008
Author Response
Manuscript ID: ijms-2350336
Dear Reviewer,
We really appreciate the time spent reviewing our work as well as all the suggestions made to the manuscript. We accepted most of the reviewer´s comments and a point-by-point report is provided. An appropriate reply to each comment is given in detail. We strongly believe that these modifications have improved the manuscript.
This manuscript describes a very interesting study dealing with transcriptome-level analysis of low-concentration exposure to 2 bisphenol A, S, and F in prostate cancer cells. While a literature review is needed to update the references, here are some references that should be included in the references:
DOI: 10.1186/s12263-017-0555-5
DOI: 10.1177/1559325815610582
We really appreciate your recommendations. This first two references were included in the introduction lines 106 and 112
DOI: 10.3390/ijms24065342
DOI: 10.3390/ijms23031216
These articles were cited in lines 405 to 408 to further explain the complexity of Bisphenol exposure endocrine disrupting mechanisms.
Reviewer 3 Report
In manuscript IJMS-2350336, Co rtés-Ramírez S. and colleagues investigated the transcriptomic effect of low dose Bisphenol A (BPA) and its analogues exposure on prostate cancer cells and identified dozens of pathways regulated. Clearly this study gave a glimpse of the big picture of transcriptomic profile change on the effect of BPA and its analogues, but need further work on molecular level study. Overall, this manuscript was well written and organized. However, there were still some minor issues need to be addressed. Here are the specific comments:
Concerns:
(1) In Figure 1D, it is difficult to understand that vehicle (EtOH) had the similar pattern (cytotoxicity and proliferation) with BPA or its analogues in proliferation assay.
(2) In Figure 2, the legends of X and Y -axis are not readable.
(3) In Figure 3 and 4, panel D was mistakenly cited as A.
(4) In Figure 6, the legend is not correct, BPA and its analogues only used at 5 nM.
(5) Lin 254, please indicate the right citation of figure.
(6) Line327-328, should be estrogen receptor 1(ESR1) and estrogen receptor 2 (ESR2).
(7) Line 340-341, the proliferation in reference 45 showed opposite result. Authors need to rewrite the discussion in this part.
(8) In the supplementary data set, the section of Overlap did not display the right format, currently showed format of date.
The English language is fine.
Author Response
Manuscript ID: ijms-2350336
Dear Reviewer,
We really appreciate the time spent reviewing our work as well as all the suggestions made to the manuscript. We accepted most of the reviewer´s comments and a point-by-point report is provided. An appropriate reply to each comment is given in detail. We strongly believe that these modifications have improved the manuscript.
Concerns:
(1) In Figure 1D, it is difficult to understand that vehicle (EtOH) had the similar pattern (cytotoxicity and proliferation) with BPA or its analogues in proliferation assay.
We appreciate your feedback and would like to address your concerns. In previous experiments in our lab we noticed that hormone-depleted cells experience a proliferation decrease, and after some hours cells recover their proliferative behavior.
(2) In Figure 2, the legends of X and Y -axis are not readable.
(3) In Figure 3 and 4, panel D was mistakenly cited as A.
Thanks for your kind comments, all figures 2-4 were improved in this version of the article.
(4) In Figure 6, the legend is not correct, BPA and its analogues only used at 5 nM.
This legend was changed by “ Figure 6. Cell cycle analysis for prostate cancer cell lines LNCaP (A) and PC-3 (B) exposed to bi-sphenols A, S and F at 5 nM. Bisphenol A was capable of inducing cell cycle arrest in PC-3 cells. * Statistically significant against vehicle control.”
(5) Lin 254, please indicate the right citation of figure.
We include the figure citation “There were twenty-three GO terms shared between three bisphenol exposures (Fig 4G)”.
(6) Line327-328, should be estrogen receptor 1(ESR1) and estrogen receptor 2 (ESR2).
It was changed by “such as estrogen receptor (ERα, ESR1) and estrogen receptor (ERβ, ESR2)”
(7) Line 340-341, the proliferation in reference 45 showed the opposite result. Authors need to rewrite the discussion in this part.
This paragraph was rewritten “Previous evidence characterized the effect of BPA exposure in human prostate cancer cells (LNCaP), and they found the transcriptional profile was involved in cell cycle regulation and proliferation [45], but they reported an increase in cell proliferation while we observed no statistically significant differences in cell proliferation for LNCaP cells. This difference can be explained due to the time difference between our studies, as they exposed LNCaP cells to bisphenols for 24 h, and we performed most of the assays after 48h exposure, considering the time to complete one cell replication cycle”.
(8) In the supplementary data set, the section of Overlap did not display the right format, currently showed format of date.
We changed the supplementary data to the proper format.
Round 2
Reviewer 1 Report
The quality of presentation has been greatly improved with figures that are now easily readable and clearer. The authors have also been responsive to several of my comments and implemented appropriate changes. I appreciate the effort made to improve this work. I think there are still some minor English editing that could be slightly tweaked to make the reading flow better but otherwise this work could be considered publishable.
Some minor English editing is recommended to make the reading flow better and some text can be "massaged" a bit.
Author Response
Dear reviewer, thank you for your kind comments.
We attended to them and we are submitting a new version of the article that addresses your recommendations
Best
